# Cochlear Implantation in a Patient with Implanted Trigeminus Stimulator—Clinical Considerations for Using Two Different Electrical Stimulators in the Same Patient and Our Results

**Daniel Polterauer [1],\* , Maike Neuling [1], Sophia Stoecklein [2] and Joachim Mueller [1]**

[1] Section Cochlear Implantation, Department of Otorhinolaryngology, University Hospital of Munich (LMU), 81377 Munich, Germany; maike.neuling@med.uni-muenchen.de (M.N.); joachim.mueller@med.uni-muenchen.de (J.M.)

[2] Department of Radiology, University Hospital of Munich (LMU), 81377 Munich, Germany; sophia.stoecklein@med.uni-muenchen.de

\* Correspondence: daniel.polterauer@med.uni-muenchen.de

**Abstract:** Implantation of two electrical stimulators of different cranial nerves in one patient is rare. We report the case of a forty-seven-year-old patient already implanted with a trigeminus nerve stimulator. In addition, this patient suffered from hearing problems. In one ear, the patient was deaf. On the other side, the patient wore a bone conduction hearing aid to improve hearing. In this complex situation, we decided to check the possibility of cochlear implantation on the deaf side. Finally, we managed to provide electrical stimulation of the auditory pathway of the deaf ear to improve the patient's hearing tests. In addition, this case report shows how the trigeminus stimulator interferes with the electrical stimulation in auditory evoked potentials measurement of the auditory brainstem and cortex via EABR (evoked auditory brainstem response) resp. EALR (evoked auditory late response).

**Keywords:** cochlear implant; EABR; EALR; trigeminal nerve stimulator

## 1. Introduction

Today, cochlear implantation in deaf patients is regularly a straightforward procedure [1]. In the case presented here, rare circumstances made it difficult to find a good solution using electrical stimulation of the auditory pathway. As well as having a bone conduction hearing aid contralateral to the deaf ear, our patient was implanted with an electrical stimulator that aims to stimulate the trigeminal nerve to minimize the patient's pain. Due to possible electrical interferences between the trigeminal nerve stimulator and the cochlear implant that was planned to be implanted, several pre-operatively checks were required. Our patient was therefore not only suffering from pain, but also from deafness on one side and hearing loss on the other ear. In this case report, we describe how cochlear implantation was made possible, how the patient performed clinical measurements, and what special issues were recognized.

## 2. Case Presentation

A forty-seven-year-old patient presented himself to our clinic asking for rehabilitation of hearing. In childhood, the patient reported a temporal bone fracture on both sides. He underwent cholesteatoma surgery in both ears. In 1997, on the right side, he received a trigeminal nerve stimulator system (stimulator + electrode) because of diagnosed trigeminal neuropathy for pain control after a skull base fracture. Since the first implantation of the trigeminal nerve stimulator system, the stimulator unit has been exchanged multiple times due to battery exhaustion. Related to his hearing situation, the patient reported deafness for about 40 years in the right ear; on the left side, the patient reported a (combined) hearing

loss for around 40 years. On the left side, he used a hearing aid until BAHA implantation. This hearing aid was still in use in 2001 when he received the BAHA. On the right ear, he never used a hearing aid. Furthermore, in February 2012, the patient was diagnosed with borreliosis and meningitis.

Diagnostics confirmed deafness of the right ear and profound hearing impairment of the left ear. This finding resulted in a WHO grade 4 of hearing impairment [2,3].

At that time, the patient had the trigeminal nerve stimulator system "ACTIVA SC/PC" of the manufacturer Medtronic implanted.

Our first hearing tests (see Figure 1) on the left ear stated an unaided 500 to 4k pure tone audiogram of 88.0 dB HL air conduction and 57.0 dB SPL bone conduction. Using the BAHA system resulted in an aided pure tone audiogram of 50.0 dB HL for 500 Hz to 4 kHz and the aided monosyllabic word understanding of 85% at 65 dB SPL in the Freiburg speech test. We did not find hearing via air or bone conduction and also did not find speech understanding in the right ear.

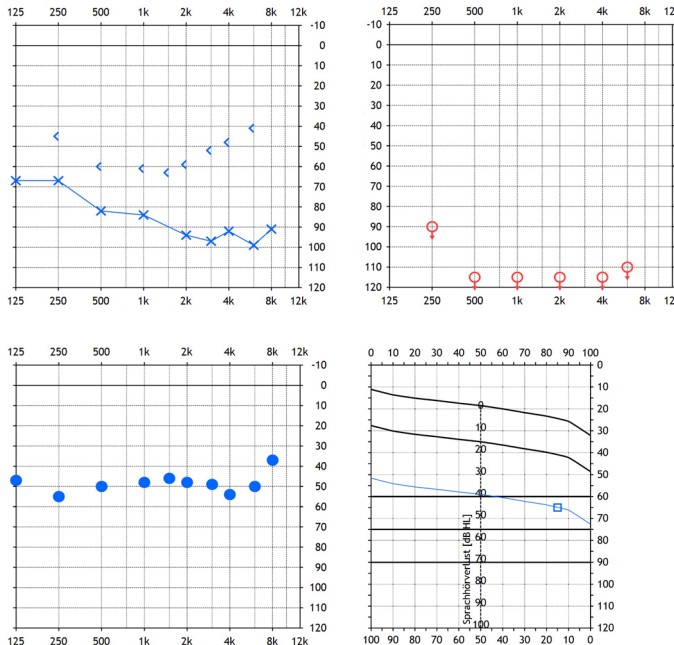

**Figure 1.** Hearing tests using tone and speech audiometry of the right and left ear at the first consultation: unaided tone-audiometry (**top left**: left ear; **top right**: right ear) and aided tone (**bottom left**) and Freiburg mono-syllable speech audiometry for the left ear (**bottom right**) using the BAHA hearing aid. (The copyright of these figures belongs to INNOFORCE Est.).

Considering cochlear implantation of the right ear, we needed to check possible electrical interactions between two electrically stimulating devices, interference, or cross-stimulation. The question was whether the existing electrical stimulation system for the trigeminal nerve would be compatible with an additional electrical stimulation pulse of a cochlear implant system for the auditory nerve. One major request from Medtronic was that the cochlear implant system had to only stimulate inside the cochlea. In CI, this means solely intracochlear contacts are used for active and reference stimulation electrodes. Such a stimulation mode is called bipolar stimulation. However, in today's cochlear implants, monopolar against bipolar stimulation mode has become the standard. In CI, monopolar stimulation mode means that intracochlear contacts are used as active stimulation electrodes and an extracochlear electrode is used as the reference stimulation electrode. The earlier stimulation modes—the old-fashioned stimulation modes—are no longer standard due to a lower chance of auditory nerve response recording, potentially lower speech understanding, and higher power consumption [4–8]. Modern cochlear implant systems either do not offer such a stimulation mode or do not use it by default.

Cochlear Ltd. offers such a bipolar stimulation mode [9]. Thus, the only compatible CI system that fulfilled the requirements according to Medtronic specifications of compatibility to be implanted legally and that was available to us with the safety considerations to the brain electrode was the CI500 series from Cochlear Ltd., Sydney, Australia. Therefore, we were able to offer cochlear implantation to the patient.

In 2018, the patient's deaf right ear was implanted with a cochlear implant offering bipolar stimuli (electrode Cochlear Profile Slim Straight (CI522) offering 22 intra-cochlear electrodes for stimulation of the auditory nerve and two reference electrodes) to enable hearing (see Figure 2).

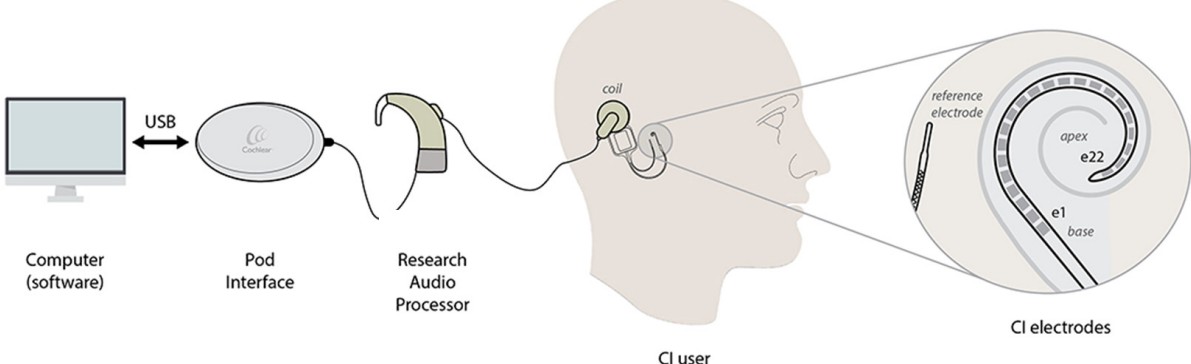

**Figure 2.** Hardware setup for CI measurements and electrode illustration ([10], Figure 1, modified text "Research Audio Processor" → "Audio Processor", licensed CC 4.0).

Standard surgery through classical access (mastoidectomy, Facial Recess Approach, cochleostomy) was uneventful despite the skull base fracture. Intra-operatively, standard tests were performed: impedance test, auditory nerve response test, and electrically evoked stapedius reflex test. First, the impedance test measurement allows the measurement of the impedances of all 22-intracochlear electrodes. The auditory nerve response test checks for compound action potentials (ECAPs) during electrical stimulation by the cochlear implant. These early responses of the auditory pathway are equivalent to electrocochleography, where acoustic stimulation evokes the compound action potential. The electrically evoked stapedius reflex (ESR) test, in contrast to the other two intra-operative standard measurements in cochlear implants, can check the auditory pathway up to the brainstem. On the other side, stapedius reflex measurement is a semi-objective test, unlike impedance and ECAP. Objective stapedius reflex detection is currently not part of the clinical routine but is evaluated in studies [11]. Stimulation is provided by the cochlear implant using a 0.5-s-long burst stimulus. The stapedius reflex path is a chain of the auditory nerve, the cochlear nucleus, the superior olivary complex, and the nucleus of the facial nerve [12].

For this patient, we were able to perform impedance and ECAP measurements. An ESR test was not possible. According to the anatomical findings, the surgeon could not visualize the stapedius reflex, as the stapes and its tendon were missing. In addition, the surgeon could not expose the stapedius muscle in this patient. Impedance and ECAP were in the normal range.

Post-operatively, we recorded an X-ray to control the position of the cochlear implant (see Figure 3). This X-ray also documents the position of the implanted trigeminus stimulator's housing, lead, and electrode. The radiologic department proved no disconnections, correct positioning of the cochlear implant, and an insertion depth of the intracochlear stimulation electrode of around 360°. At a later stage, we estimated the cochlear parameters from the pre-operative CT scan using the OTOPLAN 3 software (of diameter = 10.1 mm, height = 4.8 mm, width = 7.5 mm, and cochlear duct length = 39.6 mm using OTOPLAN 3 (compare Figure 4; according to [13]).

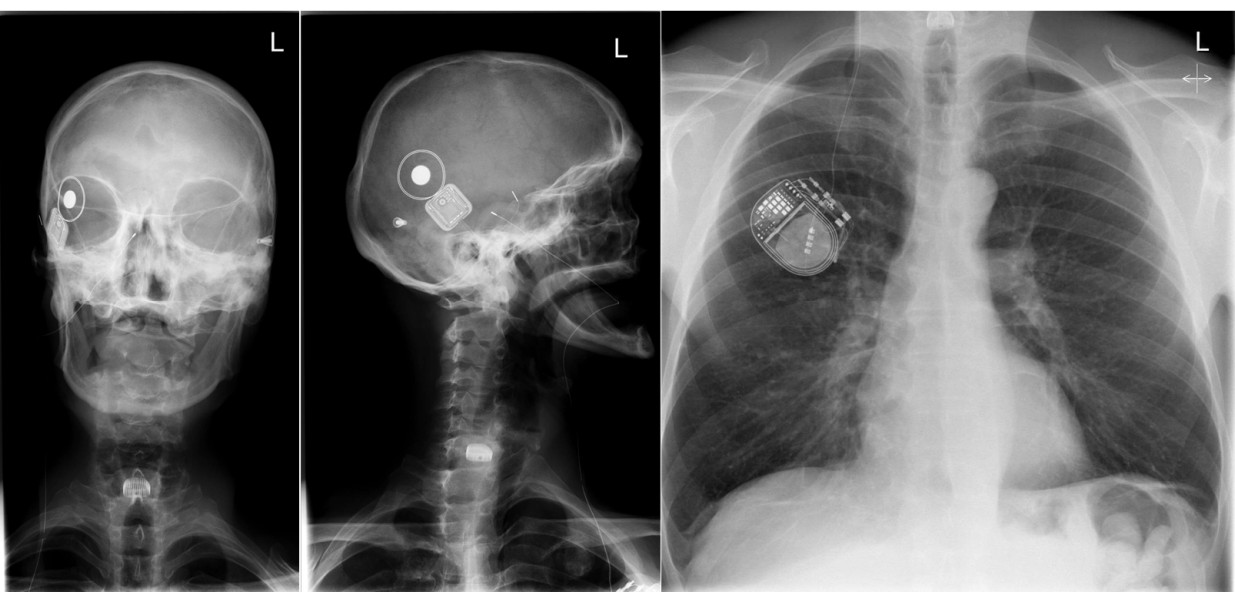

**Figure 3.** X-ray of the patient's head (**left and central subfigure**) and the patient's thorax (**right subfigure**), 3 months after surgery, for position control of the cochlear implant. Besides the cochlear implant, the stimulation electrode and the electrode lead of the trigeminus nerve stimulator are visible on the right side of this image.

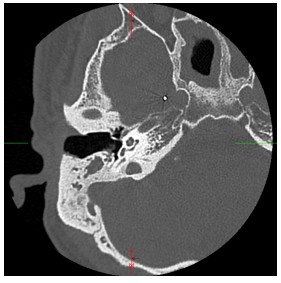 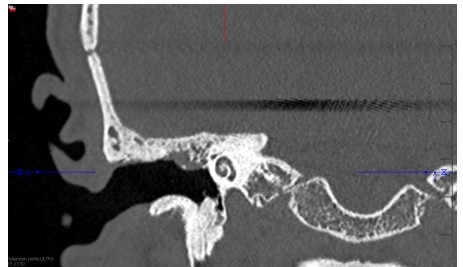 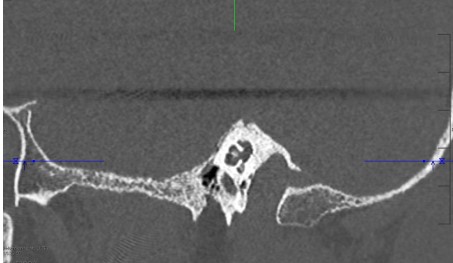

**Figure 4.** Cochlear analysis of the right ear (CI ear) pre-operatively (in axial (**left subfigure**), coronal (**central subfigure**), and sagittal view (**right subfigure**)).

For safety considerations, we proceeded with the initial activation of the cochlear implant system comparable to ABI first-fitting procedures (intermediated care unit, monitoring, and anesthesia team on standby) for being aware of a theoretical possible interference of the two neuro-stimulators running simultaneously. In addition, we invited external specialists from the manufacturer to attend since there are no reports of experience of a combination of these two devices. Thus, we performed the first activation and fitting of the cochlear implant system in our regular time window six weeks after CI surgery. According to the recommendation of the trigeminus nerve stimulator manufacturer, only the bipolar stimulation mode was selected for the cochlear implant fitting, using only intracochlear electrodes on the intracochlear electrode array. After testing different electrode configurations and distances for this stimulation mode, we found confident results for BP+3 (BP+3 means that the reference electrode is always three electrodes away from the stimulation electrode). In contrast, other BP values showed higher interferences between the CI and the trigeminal stimulator or smaller dynamic range (DR). In addition, we unsuccessfully tried to deactivate electrodes for higher DR. Using BP+3, we could provide a small DR for electrical stimulation without interference between the two neuro-stimulators, resulting in a comfortable stimulation for the patient. DR is the difference between the threshold and loud, but not uncomfortable, stimulation limits (compare [14]). The DR was around

15 CL (=manufacturer-specific current level). It can be converted to microampere using the following equation given by Cochlear [15].

$$I\,[\mu A] = 17.5 \times 100(I\,[CL]/255), \tag{1}$$

Based on the cochlear implant manufacturer's standard values for lower and upper stimulation levels, the reference DR is around 45 CL. Such a small DR is uncommon and results mostly in lower performance in speech understanding. Besides these optimizations of fitting values, we selected a relatively high pulse width of 150 µs to provide sufficient stimulation charge at low stimulation amplitude. Additionally, we selected a stimulation frequency of 250 Hz resulting from stimulation amplitude and pulse width being the maximum stimulation frequency value possible to enable maximum resolution of auditory information submitted by the CI.

One month after the initial activation of the cochlear implant system, an electrically evoked auditory brainstem response (=EABR) was successfully recorded using clinical software and hardware. Stimulation was provided via the EABR task in Cochlear Custom Sound EP 5.0. Recording was performed using the Nihon Kohden Neuropack S1 MEB9400. We chose this method as it is known to also be reliable in difficult-to-record responses, such as patients with malformations. For EABR stimulation, we selected electrode number 11 in the middle segment of the active electrode array using bipolar stimulation mode (BP+3) at a subjectively loud sensation. Due to continuously appearing strong artifacts and spontaneously appearing strong artifacts, we used 8000 instead of the usually recorded 1000 averages. This modification provided low residual noise. In contrast to pre-operative EABR recordings, no myogenic artifacts hardened the recording of waveforms [16]. As in pre-operative EABR recordings, we observed a much stronger stimulus artifact than in other patients. Apart from this setting, we used the default EABR setting from [16] for intra-operative EABR. Therefore, we applied surface recording electrodes to the contralateral mastoid (inverting), high forehead (noninverting), and lower forehead (ground). In both evoked potential devices, we set the recording window to 10 ms and the bandpass filter to 50–3000 Hz. We set the amplifier range to 500 mV, and the rejection threshold to ±150 µV, and we strictly monitored the patient's movements. Apart from the rejection threshold, the settings are standard in our department. The rejection threshold had to be increased to ±150 µV to allow recording of waveforms at all. In combination with the increased number of averages per waveform, we were able to record brainstem responses (see Figure 5). Both essential responses, eIII and eV, were securely reproduced. Latencies of wave eV are higher than in typical EABR. In this patient, we observed a value of around 5.15 ms while the reference value is 3.98 ± 0.24 ms [17]. Prolonged latencies of eV are known in cases of deprivation [18,19] or in autism [20].

In addition, electrically evoked auditory late response (= EALR) was recorded to evaluate the cortical auditory area. This measurement is important as only that way we could test the whole auditory pathway up to the auditory cortex [21]. We selected the default settings for post-operative EALR [22]. We used a band-pass filter from 0.5 Hz to 100 Hz, collecting 50. We applied rejection for sweeps with an amplitude higher than ±100 µV. Other settings were identical to those for the EABR. As with the EABR recording in the patient, we needed to increase the rejection level to a higher than standard value due to artifacts blocking any kind of waveform recording. In contrast to pre-operative EALR, we did not have to handle high CI stimulation or myogenic artifacts [23]. In this patient, continuous and spontaneous stimulation noise led to challenging recording circumstances, as seen in the EABR. Therefore, we averaged four recordings with default settings. Figure 6 documents the recorded cortical responses. All essential responses, P1, N1, P2, and N2 were securely reproduced.

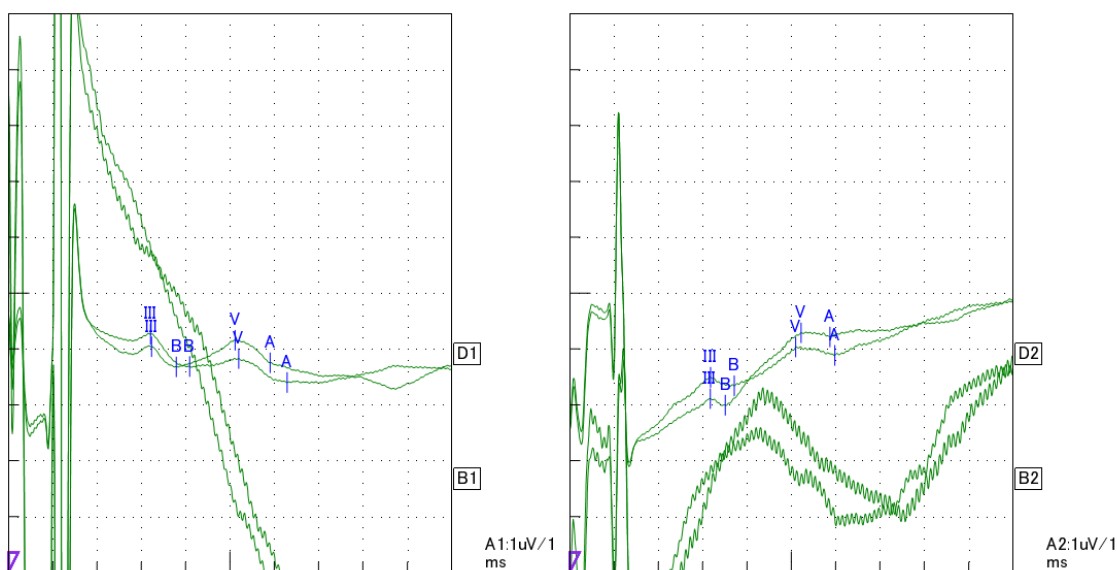

**Figure 5.** EABR at medial electrode stimulation was recorded one month after the initial activation of the cochlear implant system. The first recordings (B1/B2) failed. After setting the number of averages from 1000 (default value) to 8000 (maximum value), we could record and reproduce the brainstem responses eIII (peak marked as III; trough marked as B) and eV (peak marked as V; through marked as A). Capital letters show the recording order of these responses. 1 and 2 are the two recording channels for the right and left mastoids. Waveform illustrations by Nihon Kohden Neuropack S1 (Nihon Kohden Corporation, Tokyo, Japan) with kind permission of NIHON KOHDEN EUROPE GmbH.

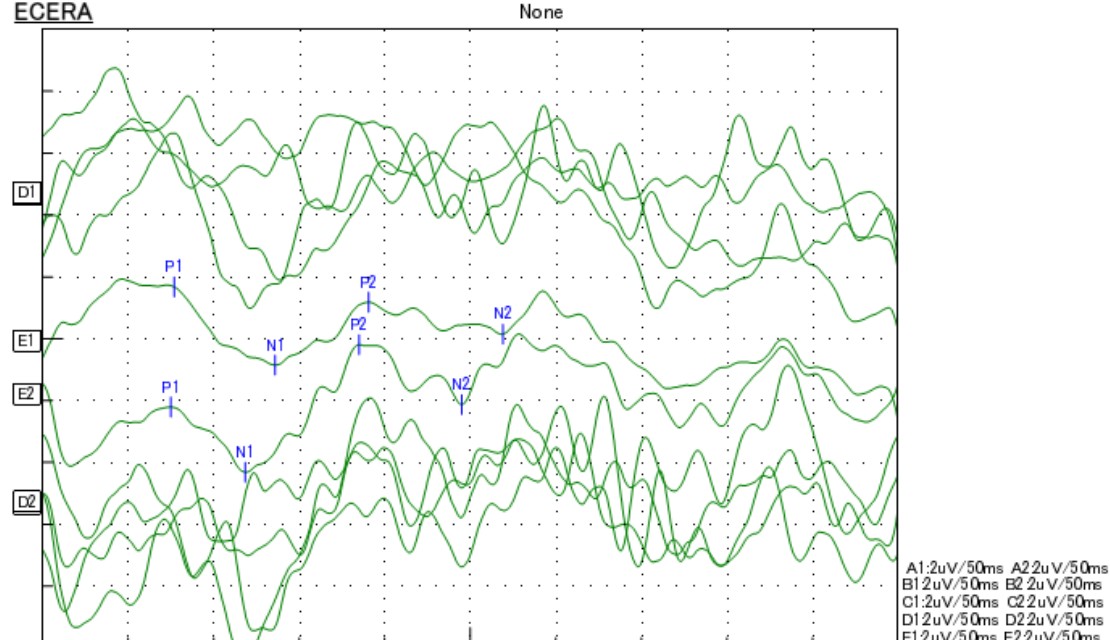

**Figure 6.** EALR at medial electrode stimulation was recorded one month after the initial activation of the cochlear implant system. The first recordings (A, B, C, and D) failed. We averaged these four recorded waveforms. In this way, we could record and reproduce the cortical responses P1–N1–P2–N2. A, B, C, D, and E show the recording order of these responses. 1 and 2 are the two recording channels for the right and left mastoids. Waveform illustrations by Nihon Kohden Neuropack S1 (Nihon Kohden Corporation, Tokyo, Japan) with kind permission of NIHON KOHDEN EUROPE GmbH.

During the follow-up appointments over the next three months, it was possible to enlarge the DR to around 20 CL after switching to a pulse width of 200 μs. However, only

two months later, DR had to be set to around 10 CL. Apart from this, the configuration of the CI stimulation values was stable in this patient and also the patient's descriptions of setting lower and upper stimulation levels for the CI electrodes were accurate and reproducible. The patient was able to differentiate steps of 1 CL which is rare from our clinical experience.

At the initial activation of the CI, audiometric tests showed an insecure threshold at 73.83 dB HL (pure-tone audiogram; mean of frequencies 500 Hz to 4 kHz) and no speech understanding in either Freiburg numbers or a monosyllables test (contralateral ear masked). The experience of CI stimulation was not securely a hearing. Two months later, the patient reported electrical stimulation in the head and no hearing via speech processor usage. In contrast, the patient reported hearing when stimulation was provided via single electrode stimulation during the CI fitting session. An additional month of CI usage later, the patient could also hear when using the speech processor and not just during single electrode stimulation. The patient reported that, in the evening, a volume increase via remote control was needed to correct the subjective volume to a comfortable level. Half a year after activation of the CI system, audiometric tests showed/confirmed a hearing threshold of 69.3 dB HL (pure-tone audiogram; mean of frequencies 500 Hz to 4 kHz). Additionally, we performed a loudness scaling using a broad-band noise signal. Results showed/confirmed a linear loudness increase from a threshold of 60 dB SPL to medium sound sensation at 80 dB SPL. The patient's speech understanding in Freiburg numbers was 0% at 65 dB SPL and 20% at 80 dB SPL (contralateral ear masked). In the Freiburg monosyllable test, no understanding was possible at any time. Three and a half years after implantation, in the Oldenburg sentence test, the patient achieved an SNR of 4.2 dB for a speech recognition threshold of 50% best aided (CI + BAHA). The patient still uses the CI system for around 13 hours a day and is satisfied with his hearing. On the contralateral ear, the patient uses the BAHA. Though the BAHA enables good speech understanding (100% at 65 dB SPL in Freiburg numbers and 75% at 65 dB SPL in Freiburg monosyllable test) the patient reported that the combination of CI and BAHA outperforms a unilateral hearing in daily life.

## 3. Discussion

The extraordinary case of cochlear implantation illustrated in this case report shows how complex it can be to check a CI indication. The existing trigeminal nerve stimulator system presented many obstacles (compare cases of CI and deep brain stimulator in one patient [24–28]). That the manufacturer of the trigeminal nerve stimulator system only allowed intracochlear stimulation by the CI system was challenging. However, we found a solution for this patient by allowing cochlear implantation using the bipolar stimulation mode instead of today's standard in CI, the monopolar stimulation mode. Luckily, we found one manufacturer, Cochlear Ltd., that still offers the option to select this old-fashioned stimulation mode. After some minor modifications in the setup (higher rejection levels and higher number of averages), the electrophysiological measurements of the auditory pathway objectively confirmed hearing in this patient (EABR and EALR), though strong interferences caused by the trigeminal nerve stimulator made this routine measurement challenging (compare [29]. As these interferences partly appeared continuous and partly a suddenly large stimulation, only a patient-specific setup can help in modifying the rejection threshold to a higher (but as low as possible) value—and in addition set the averages per waveform to the maximum level and/or average multiple averaged waveforms or raw data after the EABR/EALR. The interferences seem to be caused by the trigeminal nerve stimulator in stand-by (continuous interferences) and active stimulation of the trigeminal nerve stimulator (sudden interferences). In the audiometry results, our patient accomplished a little speech understanding half a year after the initial activation of the CI system and also benefits from wearing the CI and the contralateral BAHA simultaneously, in comparison to one device alone. In summary, this report shows that cochlear implantation can be achieved—and can improve hearing—in patients with an existing trigeminal nerve stimulator system when considering the need to modify settings in the

CI system to avoid interaction between the trigeminal nerve stimulator system and the cochlear implant system.

**Author Contributions:** Conceptualization, D.P.; methodology, D.P.; validation, D.P., M.N., S.S. and J.M.; formal analysis, D.P. investigation, D.P.; resources, D.P.; data curation, D.P.; writing—original draft preparation, D.P.; writing—review and editing, D.P., J.M. and M.N.; visualization, D.P. and S.S.; supervision, J.M.; project administration, D.P.; funding acquisition, no funding. All authors have read and agreed to the published version of the manuscript.

**Funding:** This research received no external funding.

**Institutional Review Board Statement:** Ethical review and approval were waived for this study due to solely retrospective analysis for this case report.

**Informed Consent Statement:** Patient consent was waived due to solely retrospective analysis for this case report.

**Data Availability Statement:** Data are contained within the article.

**Conflicts of Interest:** The authors declare no conflicts of interest.

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
