# Peer review of "Cochlear Implantation in a Patient with Implanted Trigeminus Stimulator—Clinical Considerations for Using Two Different Electrical Stimulators in the Same Patient and Our Results"

_2504-463X, doi:10.3390/ohbm5010002_

Round 1

Reviewer 1 Report

Comments and Suggestions for Authors

Dear authors,

the presented case report is very interesting as it deals with a rare condition. Evaluating the pros and cons of dual electrical stimulation is certainly crucial, given that a cochlear implant procedure in such a case can significantly improve the patient's quality of life.

The figures presented are intriguing, and the captions are appropriate and clear.

Here are a few suggestions: the case presentation (67-92) could be described more clearly. The contraindication in cases like the one presented regarding CI with bipolar stimulation is explained, followed by a brief description of why monopolar stimulation is currently preferable. Then, there's a return to the choice of the implant model, after indicating the Cochlear IC series compatible with the Medtronic stimulation system on the preceding page... it's a bit confusing. It would be preferable, if possible, to better organize the premises before the intervention and the subsequent described solutions.

Additionally, in lines 86/87, remove the sentence related to the abbreviation of bipolar stimulation here and introduce it when mentioned for the first time, perhaps including a brief definition of mono and bipolar stimulation. These concepts are undoubtedly familiar to many readers, but for some, it might be interesting and helpful to better understand what they entail.

The paragraphs regarding the measurements taken are detailed and clear. However, it could be beneficial to outline the instrumental exams conducted during the various follow-ups, the fitting choices, and the results obtained.

Question: How soon after the surgical intervention was the cochlear implant activated?

The discussion could be articulated a bit more effectively, perhaps highlighting more prominently the solution found to enable this patient to undergo dual electrical stimulation safely.

Author Response

Dear reviewer,

Thank you for your interest in our work and helping suggestions.

Regarding lines 67-92, following your comment, we reordered and slightly modified this part to make it easier to read and understand.

Regarding lines 86/87, we removed the definition of the abbreviation and added definitions of mono- and bipolar stimulation mode in CI. Because as you correctly commented some readers might need this information and in addition this improves readability.

Regarding the intrumental exams, we added some information about used SW and HW and also explained why we selected non-standard-values.

Regarding fitting values, we put in more details about the selected BP+3, electrode deactivation tests, pulse width selection and the stimulation frequency. Additonally, we provided more details about the follow-up sessions for CI fitting.

Regarding obtained results, we added inital results at first fitting and follow-up results also regarding the contralaterally worn BAHA.

You asked for the time of the first activation of the CI system. We performed it in our regular time window six weeks after CI surgery. We added this information also to the manuscript.

The discussion was extended as requested for more details especially regarding the solution using biploar instead of the monopolar stimulation mode for the CI.

Reviewer 2 Report

Comments and Suggestions for Authors

This study provides clear-cut information about the patient's history, the existing trigeminal nerve stimulation, and the novel intervention of cochlear implantation. The authors have extensively discussed how successful cochlear implantation, in the presence of a trigeminal nerve stimulator, adds valuable insights to the field. Here are some suggestions that can enhance the value of this study and need to be addressed:

 Provide more details on the patient's subjective experience post-cochlear implantation.

 Provide the pre-and post-operative measures taken during the study.

 Elaborate on the observed interference between the trigeminal nerve stimulator and auditory evoked potential measurements in the discussion section of the manuscript.

 Also, discuss potential mechanisms of interference and implications for future cases.

Author Response

Dear reviewer,

Thank you for your interest in our work and helful suggestions.

As requested we added more details to existing follow-ups and added some more appointments of the patient between them to go into details.

Regarding pre- and post-operative measures, we added more details including results for best aided sentence test.

To report about the interferences in EABR and EALR, we put in brief information about challenges and solutions.

In the dicussion, we added as you requested our theory of the interferences' origin as well as a short information for future cases how to deal with these interferecnes (CI fitting as well as EABR/EALR measurements).